# Plasma Membrane-Associated Restriction Factors and Their Counteraction by HIV-1 Accessory Proteins

**DOI:** 10.3390/cells8091020

**Published:** 2019-09-02

**Authors:** Peter W. Ramirez, Shilpi Sharma, Rajendra Singh, Charlotte A. Stoneham, Thomas Vollbrecht, John Guatelli

**Affiliations:** 1Department of Medicine, University of California San Diego, La Jolla, CA 92093, USA; 2VA San Diego Healthcare System, San Diego, CA 92161, USA

**Keywords:** HIV-1, Vpu, Nef, BST-2, SERINC5, CD4

## Abstract

The plasma membrane is a site of conflict between host defenses and many viruses. One aspect of this conflict is the host’s attempt to eliminate infected cells using innate and adaptive cell-mediated immune mechanisms that recognize features of the plasma membrane characteristic of viral infection. Another is the expression of plasma membrane-associated proteins, so-called restriction factors, which inhibit enveloped virions directly. HIV-1 encodes two countermeasures to these host defenses: The membrane-associated accessory proteins Vpu and Nef. In addition to inhibiting cell-mediated immune-surveillance, Vpu and Nef counteract membrane-associated restriction factors. These include BST-2, which traps newly formed virions at the plasma membrane unless counteracted by Vpu, and SERINC5, which decreases the infectivity of virions unless counteracted by Nef. Here we review key features of these two antiviral proteins, and we review Vpu and Nef, which deplete them from the plasma membrane by co-opting specific cellular proteins and pathways of membrane trafficking and protein-degradation. We also discuss other plasma membrane proteins modulated by HIV-1, particularly CD4, which, if not opposed in infected cells by Vpu and Nef, inhibits viral infectivity and increases the sensitivity of the viral envelope glycoprotein to host immunity.

## 1. Introduction

The host–pathogen relationship typically involves a conflict between host-defenses and viral countermeasures, which over time and consequent to cross-species transmission often leaves telltale genetic signatures [1]. Much of this conflict, particularly in the case of enveloped viruses, plays out on the plasma membrane. One aspect involves an effort by the host to kill infected cells before they substantially amplify the virus. This cell-mediated immune surveillance can be mediated by cytotoxic T lymphocytes (CTL), which recognize foreign viral peptides displayed at the cell surface by class I MHC molecules [2], or by natural killer (NK) cells, which recognize abnormalities in the plasma membrane as well as viral envelope glycoproteins such as HIV-1 Env once on the cell surface and bound by specific antibodies [3]. Moreover, the host cell can express, either constitutively or in response to interferons, various proteins with direct antiviral activity, some of which act on HIV-1 virions as they assemble and bud from the plasma membrane [4,5,6,7].

HIV-1 encodes two membrane-associated “accessory” proteins to counteract these host defenses: Nef and Vpu down-regulate class I MHC, inhibiting antigen presentation and CTL activity [8,9]; they down-regulate NK receptors [10,11]; and they down-regulate cell-intrinsic host antiviral proteins, so-called restriction factors, that directly inhibit the release or infectivity of progeny virions [4,5,6,7]. The best characterized of these restriction factors are BST-2, which traps budded virions on the surface of infected cells and is counteracted by Vpu [4,5], and SERINC3 and SERINC5, which inhibit the infectivity of progeny virions and are counteracted by Nef [6,7]. These antiviral host proteins were discovered through quests to explain in vitro virologic phenotypes: The efficient release of cell-free virions from most cell types requires *vpu* [12,13], whereas optimal virion-infectivity requires *nef* [14]. Over two decades of detailed studies have characterized the Vpu and Nef proteins: Their timing of expression, their diverse activities and cellular targets, their structures, and their mechanisms of action. These mechanisms center on their abilities to act as non-enzymatic adaptors that recruit cellular components of the membrane protein quality control and trafficking machinery to their targets. The result is the depletion of proteins from the plasma membrane that are deleterious to the virus. Remarkably, the targets of Vpu and Nef include CD4, the virus’s primary receptor [15,16]. Here, we review the key plasma-membrane associated restriction factors, BST-2 and the SERINC proteins, as well as the antiviral effects of CD4 during virion-production. We also review Vpu and Nef, including their cellular cofactors and the key protein-protein interaction interfaces through which these viral proteins act as membrane-associated adaptors. Finally, we introduce the plethora of changes to the plasma membrane recently cataloged, particularly by high-depth proteomic analyses, and how the significance of these changes might be assessed.

## 2. The Key Plasma Membrane Proteins that Inhibit HIV-1 Release and/or Infectivity and How They Work

### 2.1. BST2: Historical Basis of Discovery (The Inhibitor That Vpu Counteracts to Enhance Virion-Release); Protein Topology; Mechanism of Action

BST-2 (bone marrow stromal antigen-2) is constitutively expressed in many cell types including the lymphoid and myeloid cells that host HIV-1 in vivo [17], but like other classic restriction factors it is interferon-inducible [18]. It is a potent restrictor of several families of enveloped viruses that assemble at the plasma membrane, including retroviruses, filoviruses, γ-herpesviruses, and arenaviruses [4,5,19,20,21]. BST-2 has several aliases including Tetherin, a renaming of the protein based on its ability to trap or “tether” budded virions on the surface of the cell that produced them; PDCA-1, plasmacytoid dendritic cell antigen-1 (a prominent surface protein on these cells but not specific to them); and CD317. BST-2 is a dimeric type II transmembrane glycoprotein that associates with lipid rafts. BST-2’s topology as a transmembrane protein is unusual in that its C-terminal end is modified by a glycosyl-phosphatidylinositol (GPI) anchor (Figure 1) [22]. This topology enables BST-2 to insert one membrane anchor—usually the transmembrane domain—in the plasma membrane, while inserting the other—usually the GPI anchor—in the lipid envelope of the virion [23]. Between these two membrane anchors, the ectodomain of the BST-2 dimer forms a disulfide-linked, parallel coiled-coil [24]. This presumably rigid linear structure enables BST-2 to partition one end in the plasma membrane and the other in the virion during the budding process, physically linking the virion to the cell surface and preventing its release. These key features—a transmembrane domain, a coiled-coil ectodomain, and a GPI anchor—are necessary and sufficient for virion-trapping [25]. The identification of BST-2 as a restriction factor solved a long-standing virologic mystery: How did Vpu stimulate the release of HIV-1 virions [13]? The answer was by antagonizing BST-2 [4,5]. To do this, the Vpu proteins of group M HIV-1 bind BST-2 via a direct interaction between the transmembrane domains of each protein (see below and Figure 4) [26,27,28]. Vpu also utilizes sequences in the N-terminal region of the cytoplasmic domain of BST-2 (as well as sequences in its own cytoplasmic domain) to remove BST-2 from the plasma membrane and ultimately degrade it (Figures 3 and 6). This region of BST-2 contains putative ubiquitin acceptor sites—sequence STS—and a clathrin adaptor protein (AP) binding motif—sequence YxYxxV (Figure 6)—each of which contributes to susceptibility to Vpu [29,30]. Notably, the mRNA of BST-2 encodes an internal start codon, the use of which yields a short isoform missing the N-terminal 12 cytoplasmic residues including the STS and YxYxxV sequences [31]. Because it lacks the YxYxxV endocytosis-motif, this short isoform is expressed at higher levels on the cell surface than the long isoform, and it is a more potent inhibitor of virion-release. The short isoform is also relatively refractory to modulation by Vpu. These characterizations of the short and long isoforms are based on mutational analyses; the extent to which mixed dimers exist and their functional attributes are unclear.

Virion-entrapment by BST-2 has immunologic consequences. To the extent that entrapped virions remain on the cell surface, they increase the surface-concentration of the viral Env glycoprotein, rendering the cells more susceptible to immune surveillance by antibody dependent cellular cytotoxicity (ADCC, discussed further below in the context of CD4) [32,33]. On the other hand, to the extent that entrapped virions are internalized into endosomes, they might support endosomal sensing of infection or the presentation of viral antigens by MHC [34,35]. Moreover, BST-2 activates NF-κB, and, at least in the case of retroviruses, this activity is augmented by virion-entrapment [36,37]. This property supports a role for BST-2 as a virus-sensor. The activation of NF-κB requires tyrosine 6 but not tyrosine 8 of the YxYxxV sequence, suggesting that the endocytosis of BST-2 is not required for signaling activity. Lastly, BST-2 is a ligand for immunoglobulin-like transcript 7 (ILT-7), a receptor on plasmacytoid dendritic cells (pDCs) that negatively regulates the production of type I interferons in response to signaling through TLRs [38,39]. This immune evasion activity might explain the presence of BST-2 on many cancer cell types. Despite causing the net removal of BST-2 from the cell surface, Vpu enables HIV-1 infected T cells to inhibit the production of interferons by pDCs in a manner dependent on BST-2 [40]. It apparently does this by directing a subpopulation of residual BST-2 away from virion-assembly sites, where it remains free to interact directly with ILT-7. BST-2 is found not only on the plasma membranes but also in endosomes. In macrophages, it co-localizes with virions in large virion-containing compartments, but the significance of this is unclear [41].

### 2.2. SERINC Family Members: Historical Basis of Discovery (The Inhibitors That Nef Counteracts to Enhance Virion-Infectivity); Protein Topology; Mechanism of Action

SERINC (SERine INCorporator) proteins are named for their apparent biochemical activity—facilitating the incorporation of serines into membrane lipids [42]. These proteins are conserved from yeast to humans; lower eukaryotes encode fewer genes (one in yeast) than higher eukaryotes (five in humans). SERINC homologues share a similar topology: They are integral membrane proteins with 9 to 11 membrane spanning regions. SERINC mRNA is expressed in many tissues. Up-regulation of SERINC5 mRNA in oligodendrocytes during active myelination suggests an important role for these proteins in neuronal development and function [43,44], consistent with their suggested role in the synthesis of serine-containing lipids such as sphingomyelin and phosphatidylserine [42]. Two SERINC family members—SERINC3 and SERINC5—were identified as inhibitors of HIV-1 infectivity that are antagonized by Nef [6,7]. This identification solved a long-standing mystery: Virions of *nef*-encoding HIV-1 are more infectious than virions of isogenic viruses lacking *nef* [14]. Of the five SERINC proteins encoded in the human genome, all but SERINC2 can potentially restrict the infectivity of HIV-1, but SERINC5 is the most potent, and it appears to account for the majority of the Nef phenotype [7,45,46]. SERINC proteins localize at least partly to the plasma membrane as well as the endoplasmic reticulum, and they are incorporated into budding virions. HIV-1 Nef, as well as GlycoGag of MLV and S2 of EIAV, counteract the inhibitory effect of the SERINC proteins and restore the infectivity of virions [6,7,47]. These viral proteins decrease the expression of SERINCs at the plasma membrane and exclude them from virions. These effects appear to be mediated by clathrin and AP-2-dependent endocytosis [48,49]. Whereas the ability to inhibit HIV-1 infectivity is conserved across SERINC5 orthologs from several species, their sensitivity to Nef is not [50]; this enabled mapping of Nef-responsiveness to a specific intracellular loop of SERINC5 (see below). Unlike other restriction factors, SERINC5 is not interferon-inducible [7], and on genetic grounds it does not seem engaged in an evolutionary arms race with pathogens like HIV-1 (discussed further below) [51]. Five alternatively spliced mRNA isoforms of SERINC5 are described; the longest isoform is the most abundant, stable, and functionally relevant [52]. SERINC5 is an N-glycosylated, type II transmembrane protein with 10 membrane spanning domains, five extracellular loops, and four intracellular loops [45,53]. N-glycans and the tenth transmembrane domain along with the C-terminal cytoplasmic domain support the steady-state expression of SERINC5 [52,53]. The intracellular loop 4 (ICL4) of SERINC5 is critical for sensitivity to Nef: Hydrophobic residues in the human protein—L350 and I352—are required [50]. ICL4 also contains a CCFCS sequence, which is a putative palmitoylation site [54] and an EDTEE sequence, which can bind the µ subunits of the clathrin adaptors AP-1 and AP-2 in vitro and appears to confer partial resistance to Nef [55].

How SERINC5 inhibits the infectivity of HIV-1 virions is an open question. An important mechanistic clue is that SERINC5’s activity depends on the envelope glycoprotein of the virus. Pseudotyping of HIV-1 with the envelope glycoproteins of VSV or Ebola virus confers resistance to the SERINCs [6]. Moreover, HIV-1 Envs differ in sensitivity to SERINC5: Envs that are generally more sensitive to neutralization by antibodies and more likely to form relatively “open” trimers are more sensitive to SERINC5 [7,56]. One model is that SERINC5 interacts with Env trimers and inhibits their activity. This model is indirectly supported by the observation that SERINC5 increases the susceptibility of HIV-1 Env to certain antibodies—primarily those targeting the MPER of gp41 but also one glycan-specific antibody—suggesting a change in the conformation of Env [57,58]. Alternatively or in addition, virion-incorporated SERINC5 might delay fusion of the virion and target cell membranes, such that the infectivity of virions with relatively unstable trimers decays [59]. SERINC proteins could plausibly modify virion-lipids in a manner that inhibits fusion, but such modifications have not been detected [60]. Moreover, inhibition of virion-fusion, at least the initiation of pore-formation detected in common assays, does not seem quantitatively sufficient to account for the inhibition of infectivity [6,7,59]. Although we have focused here on SERINC5, other cell-specific factors might contribute to the virologic phenotype of *nef*, raising the possibility that restriction factor(s) antagonized by Nef remain to be discovered [61].

### 2.3. CD4: The Primary Receptor, But Whose Interaction with Env During Virion-Production Inhibits Infectivity and Prematurely Exposes Epitopes on Env That Are Targets of Adaptive Immunity

CD4 is a dimeric type I transmembrane glycoprotein that is palmitoylated and associates with lipid rafts. CD4 is the primary receptor for HIV-1 and interacts with virion-associated Env to initiate infection of target cells [62], but in cells producing the virus it has the potential to inhibit virion-release and infectivity and to induce the exposure of epitopes on Env that render infected cells more susceptible to immune surveillance [32,63,64,65]. Env consists of two glycosylated subunits, gp120 and gp41, that are produced from cleavage of the precursor protein gp160 by Furin and Furin-like proteases within the Golgi [66,67]. Mature Env is a trimer consisting of three pairs of gp120/gp41; it localizes to virions and the plasma membrane [68,69]. gp120 is extracellular and binds CD4 and the HIV coreceptor (CCR5 or CXCR4) [70,71]. gp41 is a transmembrane protein that anchors the Env complex in the membrane and enables fusion of virions with target cells [72]. If unopposed by Vpu in the ER, CD4 can form complexes with gp160, blocking the processing of gp160 to gp120 and gp41 required for infectivity [73].

By reducing the level of CD4 at the plasma membrane, Nef and Vpu prevent deleterious consequences of CD4 on viral production, infectivity, and immune surveillance. Re-infection of already infected cells is prevented, which can trigger cell-death by apoptosis and presumably decrease virion-yield [74]. If unopposed by Vpu and Nef, CD4 inhibits virion-release by binding Env at the plasma membrane, and it accumulates in virions, reducing infectivity [63,64,75]. Prevention of CD4-Env interactions by Vpu and Nef also provides a defense against a key component of cell-mediated immunity: Antibody-dependent cellular cytotoxicity (ADCC). ADCC results from the lysis of infected cells by natural killer (NK) cells and other effector cells that recognize the Fc region of antibodies bound to the cell-surface [76,77]. Env is the only viral-specific antigen exposed on the cell-surface. If Env encounters CD4 in the infected, virion-producer cell, then conformational changes expose epitopes on Env that are otherwise only briefly exposed on virions during the infection of target cells. This epitope-exposure renders the infected cell recognizable by a wider range of antibodies, enhancing immune surveillance by ADCC [32,65,78]. ADCC has been proposed as the mechanism underlying the modest protective effect observed in the RV144 Thai HIV-1 vaccine trial, so these effects are potentially significant in vivo [79,80]. In addition to sensitizing infected cells to ADCC, the conformational changes in Env induced by CD4 sensitize Env to SERINC5; this is particularly striking for Envs that are relatively resistant to neutralization by antibodies and form relatively “closed” trimers, which comprise the majority of wild type, patient-derived proteins [81].

## 3. Vpu, Nef, and the Cell Biologic Mechanisms behind Their Modulation of Membrane-Protein Trafficking

### 3.1. Vpu: Protein Topology, Key Interactive Surfaces and Partners (Cullin1 E3 Ubiquitin Ligase; Clathrin Adaptor Protein (AP) Complexes), Subcellular Localization, and Cellular Protein Targets

Vpu is a small, type-I transmembrane protein expressed coordinately with the viral Env protein from a bicistronic mRNA [82]. Vpu consists of a minimal lumenal (extracellular) domain, a single transmembrane helical domain, and a relatively small cytoplasmic domain (Figure 2). The cytoplasmic domain, when unbound by any cellular partner, contains two α-helices connected by an acidic linker region. Vpu was often depicted with its N-terminal cytoplasmic helix along the membrane, but more recent data suggest instead that the more C-terminal cytoplasmic helix lies along the lipid bilayer [83], perhaps inserting its C-terminus into the lipid using residues such as tryptophan [84]. Vpu can oligomerize, forming an ion channel, but the functional relevance of this is unclear [85,86]. The acidic linker region between the two cytoplasmic helices of Vpu contains two serines that are phosphorylated by casein kinase II [87,88]. This phosphoserine-acidic cluster (or PSAC) supports the interaction of Vpu with two distinct cellular complexes that act as co-factors: A β-TrCP containing, cullin-1-based E3 ubiquitin ligase complex [89], and the medium (µ) subunits of the hetero-tetrameric clathrin adaptor complexes AP-1 and AP-2 [90,91]. Vpu also contains within the C-terminal region of its cytoplasmic domains an acidic leucine-based motif that directs the interaction of Vpu with AP-1 via a mechanism distinct from the PSAC-µ interaction, using different AP subunits (γ and σ1; see below, Figure 6) [30,92]. The reason for this bimodal interaction of Vpu with clathrin adaptors is unclear, but in principle it could relate to the modulation of different target cellular proteins. Functionally, the recruitment of the cullin-1-E3 ligase by Vpu results in the ubiquitination of cellular targets such as CD4 and BST-2, stimulating their degradation in proteasomes or lysosomes [29,89,93,94]. The recruitment of clathrin AP complexes presumably supports endocytosis and post-endocytic sorting of Vpu and its bound targets to lysosomes. But how are cellular targets identified by Vpu? Perhaps surprisingly, many of the interactions of Vpu with cellular targets seem based on interactions mediated by the proteins’ transmembrane domains (TMDs). This is best characterized in the case of BST-2 and Vpu: an anti-parallel interaction occurs between an alanine-face of Vpu’s TMD helix and a face of BST-2’s TMD helix that displays bulky hydrophobic side chains (Figure 4) [27]. How this seemingly bland face of the Vpu TMD specifically selects cellular targets such as NTB-A [10], PVR [95], and CCR7 [96] in addition to BST-2 is an open question [97], but modulation of these proteins requires the primary sequence of the Vpu TMD. In contrast, the modulation of CD1d by Vpu does not [98,99], and the modulation of HLA-C by Vpu requires bulky hydrophobic residues in the TMD rather than the alanines [9,100]. As a Vpu-target, CD4 seems to be an exception, with sequences in both the transmembrane and cytoplasmic domains of both proteins important for a functional interaction [101,102,103]. Vpu is distributed throughout cellular membranes including the ER, Golgi and trans-Golgi network (TGN), plasma membrane, and endosomes [104]. While concentrated at steady state in juxta-nuclear endosomes including the TGN [105], Vpu doubtless circulates throughput these membrane systems, facilitating interaction with its cellular targets. A list of these targets is shown in Table 1.

### 3.2. Nef: Protein Topology, Key Interactive Surfaces and Partners (AP Complexes), Subcellular Localization, and Cellular Protein Targets

Nef is a small peripheral membrane protein expressed early during the viral replication cycle, before the viral enzymes and structural proteins including Env. Nef, like Vpu, acts as an adaptor to recruit cellular membrane trafficking machinery to its targets (Figure 2). Nef associates with membranes via N-terminal myristoylation [16]. The N-terminus of Nef is itself relatively unstructured, although it contains a short α-helix and folds back onto the Nef-core in one crystal structure [106]. The Nef core is formed by a polyproline helix that binds the SH3-domains of certain Src-family kinases (Lck and Hck); two antiparallel helixes that at one end complete the SH3-binding domain and at the other end form a hydrophobic pocket (discussed further below); and four anti-parallel β-strands [107]. A C-terminal loop extends from the β-strand network; this loop is disordered unless Nef is bound to a clathrin AP complex, such as AP-2. When bound to AP-2, the loop becomes structured, and a canonical acidic leucine-based AP-binding motif within it interacts with the α and σ2 subunits (Figure 6) [108]. This leucine-based motif is essential for the Nef-mediated modulation of CD4 and the SERINC proteins [6,109], but it is dispensable for the modulation of class I MHC [110]. Only in the case of class I MHC has the interaction of Nef with its target been fully elucidated (Figure 5) [106]. The cytoplasmic domain of the class I α chain is sandwiched between Nef and the µ subunit of AP-1 (µ1), held along the Nef polyproline helix by several electrostatic and hydrogen bonding interactions involving both Nef and µ1. Moreover, the cytoplasmic domain of MHC-I acts as if it contains a canonical tyrosine-based µ-binding motif, inserting a tyrosine residue into the tyrosine-binding pocket on µ1. This interaction is essential to formation of the complex, and it explains why HLA-A and -B are modulated by Nef, whereas HLA-C, which lacks the key tyrosine, is not [111,112]. Thus, Nef facilitates an interaction of the MHC-I cytoplasmic domain with AP-1. In contrast, how CD4 interacts with Nef and AP complexes is an open question. Remarkably, the cytoplasmic domain of CD4 itself contains a leucine-based motif, and when serine residues upstream of the leucines are phosphorylated (for example, following T cell activation), the CD4 leucine-motif interacts with the same binding site in AP-2 that the Nef leucine-motif utilizes [113]. Presuming that the Nef leucine motif occupies this site during CD4-modulation, where does CD4 bind? One possibility is a hydrophobic patch that could form when the N-terminus of Nef folds back upon the core [114]. Another is the unfilled hydrophobic pocket between the Nef α-helices noted above. Notably, the residues within the cytoplasmic domain of CD4 and in the cytoplasmic loop of SERINC5 required for modulation by Nef are hydrophobic, suggesting that a hydrophobic interface on Nef binds these and possibly other cellular targets. Complicating the identification of such a binding site is the finding that Nef dimerizes via an interface whose residues are largely hydrophobic and are important for function [115]. Like Vpu, Nef is found throughout cellular membrane systems including juxta-nuclear endosomes and the plasma membrane. Its ability to interact with multiple members of the AP complex family of clathrin adaptors (AP-1, -2, and -3) [116,117] indicates involvement in endocytic and post-endocytic sorting events that lead to depletion of host proteins from the plasma membrane and their degradation in lysosomes. A list of Nef-targets is shown in Table 1.

### 3.3. The Cellular Pathways Co-Opted by Vpu and Nef

Nef and Vpu co-opt membrane trafficking and degradative processes to counteract their cellular targets (summarized in Figure 3).

#### 3.3.1. ERAD: Co-Opted by Vpu to Degrade CD4

ERAD (endoplasmic reticulum (ER)-associated protein degradation) is a quality-control mechanism that targets misfolded proteins in the ER for ubiquitination, translocation to the cytoplasm, and degradation by the proteasome. Vpu interacts with newly synthesized CD4 in the ER and targets it to an ERAD-like pathway [119]. Vpu retains CD4 in the ER through transmembrane domain interactions; this requires the membrane anchoring tryptophan in Vpu discussed above [120]. Vpu also interacts with the cytoplasmic domain of CD4 via α-helices in each of the proteins’ cytoplasmic domains [102,121,122]. Vpu simultaneously interacts with β-TrCP, a substrate adaptor for a Skp1-Cullin-1-F-box-containing (SCF) E3 ubiquitin ligase complex [89]. β-TrCP recognizes the DSGxxS phosphoserine-acidic cluster (PSAC) motif in Vpu. This motif is also known as a “phosphodegron” due to the presence of similar sequences in cellular proteins that interact with β-TrCP and are ultimately ubiquitinated and degraded. By recruiting this E3 ligase complex to CD4, Vpu induces poly-ubiquitination of the CD4 cytoplasmic domain on lysine, serine, and threonine residues [94]. The poly-ubiquitinated CD4 is recognized by the VCP-UFD1L-NPL4 dislocase complex, a late stage component of the ERAD pathway, which mediates the extraction of CD4 from the ER membrane to the cytosol and subsequent proteasomal degradation [94].

#### 3.3.2. Endocytosis: Co-Opted by Nef to Remove CD4 and SERINCs from the Plasma Membrane

Nef co-opts clathrin-mediated endocytosis and the adaptor protein AP-2 to target CD4 and SERINC5 away from the plasma membrane to late endosomes and ultimately to lysosomal degradation [6,117,123,124]. The structural bases of these effects are reviewed below; they are incompletely defined but presumably involve a ternary complex between Nef, AP-2, and cytoplasmic regions of CD4 or SERINC5. The rate of internalization of CD4 and SERINC5 from the plasma membrane is stimulated by Nef [124,125], and this depends on clathrin, AP-2, and AP-2 associated proteins such as Eps15 [117,123]. The net result is depletion of these proteins from the plasma membrane.

#### 3.3.3. Endo-Lysosomal Degradation

##### Pathways and Cofactors Co-Opted by Vpu to Degrade BST-2 (Ubiquitination, ESCRT- and AP-Complexes)

Clathrin adaptors and ESCRT (endosomal sorting complexes required for transport) complexes, among other membrane coat proteins and complexes, mediate protein sorting and vesicular trafficking steps required to deliver membrane proteins to lysosomes for degradation. Vpu-mediated endo-lysosomal degradation of BST-2 illustrates the challenge of integrating these processes into a compelling mechanistic model. The removal of BST-2 from the cell surface and the net degradation of BST-2 by Vpu involves, to various degrees, clathrin, AP complexes (AP-1 and AP-2), the β-TrCP/SCF ubiquitin ligase complex, and the ESCRT-0 complex component Hrs, a ubiquitin binding protein and monomeric clathrin adaptor [90,91,93,126,127]. As noted above, the cytoplasmic domains of Vpu and BST-2 contain linear motifs that bind AP complexes, and the cytoplasmic domain of BST-2 contains potential ubiquitin acceptor sites. Plausibly, the AP-binding motif in BST-2 supports constitutive endocytosis so that BST-2 reaches early, sorting endosomes.

At that junction, Vpu would recruit the β-TrCP/SCF ubiquitin ligase complex, inducing ubiquitination of BST-2 and causing it to interact with Hrs. This would divert BST-2 from a pathway of recycling to the plasma membrane and to a pathway of ESCRT-mediated degradation. What roles would the AP binding motifs in Vpu play in this scenario? They could support the trafficking of Vpu itself, or as discussed below, they could increase the affinity or change the specificity of BST-2’s interactions with AP complexes when Vpu and BST-2 are bound to each other. On the other hand, AP complex interactions might be sufficient for the net removal of BST-2 from the cell surface: The β-TrCP/SCF ubiquitin ligase complex does not seem required for this effect of Vpu [91,128,129].

##### Pathways and Cofactors Co-Opted by Nef to Degrade CD4 (AP1γ2, ALIX, β-COP)

Similarly to Vpu, Nef co-opts endosomal sorting machinery to degrade CD4 following endocytosis. Specific proteins and complexes co-opted by Nef include AP1γ2, an isoform of AP-1 [130]; β-COP, a component of COPI coats [131,132], and ALIX, an ESCRT-related protein that supports the budding of membrane into the lumen of multi-vesicular bodies [133]. Exactly how these cofactors and pathways work together to support Nef activity remains incompletely defined.

#### 3.3.4. TGN-Retention and Block to Recycling

Vpu and Nef each seem to block the recycling of their targets to the plasma membrane [98,134,135,136]. As noted above in the case of Vpu, this effect could be a consequence of the diversion of targets at the sorting endosome to degradation-pathways at the expense of recycling-pathways. In the case of Vpu, newly synthesized BST-2 is also partly retained in the TGN [135,137]. The mechanism of this Vpu-mediated block to trafficking along the biosynthetic/exocytic pathway is unknown.

#### 3.3.5. Movement within Domains of the Plasma Membrane

Vpu displaces BST-2 from virion-assembly sites within the plane of the plasma membrane [84,138]. This contributes to its activity in counteracting virion-entrapment by BST-2. It also frees BST-2 to interact with ILT-7, as noted above [40]. The displacement effect does not require the PSAC motif of Vpu, but it does involve C-terminal residues, including the leucine-based AP binding motif [84,138,139]. Exactly which domains of the plasma membrane are involved is unknown; one possibility is the Vpu moves BST-2 from lipid rafts [140]—sites of HIV-1 budding—into clathrin-coated membrane domains [141], perhaps as a first step in endocytosis.

## 4. The Structural Basis of These Cell Biologic Effects: The Interactions of Vpu and Nef with Targets and Co-Factors

### 4.1. Nef and Vpu with CD4

As noted above, the interaction of Env with CD4 has deleterious effects on virion production, infectivity, and immune surveillance, all of which are mitigated by Nef and Vpu. Vpu removes CD4 from the ER, ultimately leading to degradation of CD4 via the proteasome. Nef induces the endocytosis of CD4, ultimately leading to degradation of CD4 via the lysosome (see above). Together, these processes prevent Env from encountering CD4 either within the biosynthetic pathway or at the plasma membrane. Vpu and Nef recognize CD4 differently. Nef recognizes the cytoplasmic domain of CD4, which is partly α-helical and contains the key sequence SQIKRLL [142]. When the serine is phosphorylated, this sequence behaves as if it were an acidic leucine-based motif and binds the AP-2 clathrin adaptor, causing the endocytosis of CD4. Nef stimulates the endocytosis of CD4 independently of this serine [143]. Nonetheless, endocytosis mediated by Nef requires the isoleucine and leucine residues in the SDIKRLL sequence [125]. These residues presumably bind a hydrophobic region on Nef. As noted above, candidate regions for the interacting surface on Nef for CD4 include a hydrophobic patch formed when the Nef N-terminus folds back upon the Nef-core and a hydrophobic crevice formed between the two anti-parallel helices of the Nef-core [114,144]. Unlike Nef, Vpu seems to interact at least partly with the transmembrane domain of CD4, an interaction that requires the membrane anchoring tryptophan in the Vpu transmembrane domain described above [120]. In addition to the interaction between their transmembrane domains, the cytoplasmic domain of CD4 interacts directly with the cytoplasmic domain of Vpu [101,102,103]. The helical nature of the CD4 cytoplasmic domain seems required [101]. The specific residues needed within the helix are ill-defined, but the isoleucine and leucine residues required for recognition by Nef are dispensable. Notwithstanding all of this information, the precise structural bases of the CD4/Nef and CD4/Vpu interactions remain unsolved problems.

### 4.2. Vpu with BST-2

The interaction between Vpu and its target BST-2 is mediated by the TMDs of the two proteins. The interaction between these two α-helices occurs in an anti-parallel orientation and involves an “alanine face” of Vpu that is well conserved among pandemic (Group M) HIV-1 isolates (Figure 4) [27]. The alanines of Vpu fit into ridges formed by bulky, hydrophobic residues in the TMD of BST-2; e.g., L37 of BST-2 seems to fit into the crevice between A14 and A18 of Vpu. An invariant tryptophan residue in Vpu is also critical for this interaction; this residue probably contributes by inserting its side chain within the apolar/polar interface at the cytoplasmic face of the membrane and positioning Vpu correctly within the lipid bilayer. Certain differences in the TMD of non-human primate BST-2 relative to the human protein render HIV-1 Vpu inactive as a simian BST-2 antagonist [145]; these differences change the tilt angle of the BST-2 TMD in the membrane in a manner that disrupts its interaction with Vpu [27].

### 4.3. Vpu with β-TrCP

As noted above, Vpu binds β-TrCP, an F-box protein and substrate-adaptor for an SCF (Skp1-cullin-F-box) multi-subunit E3 ubiquitin ligase complex [89]. Recruitment of this E3 ligase complex by Vpu induces the ubiquitination of targets such as CD4 and BST-2. β-TrCP binds Vpu via its C-terminal domain, which contains WD repeats that form a β-propeller [146]. Vpu binds to β-TrCP via its DpSGxxpS motif (“pS” indicates a phosphoserine). As noted above, variations of this motif are found in cellular proteins that are substrates of the β-TrCP/SCF E3 ligase, including β-catenin. While a high resolution structural depiction of the interaction between Vpu and β-TrCP is not available, it might be partially modeled by homology with the crystal structure of the WD repeat region of β-TrCP bound to a β-catenin peptide [146]. According to NMR data, phosphorylation of the serines in the DSGxxS sequence enables electrostatic and hydrogen bonding interactions between Vpu and β-TrCP [147]. Vpu residues located upstream of the DpSGxxpS sequence (sequence LIER) also support the binding of Vpu to β-TrCP.

### 4.4. Nef and Vpu with Clathrin AP-Complexes

Nef and Vpu re-route cellular proteins by forming ternary complexes with the endosomal trafficking machinery. To co-opt that machinery, they display typical protein sorting motifs that mimic those in cellular proteins. They also take advantage of sorting motifs when present in their cellular targets. Both Nef and Vpu interact directly with members of the family of hetero-tetrameric clathrin adaptor protein (AP) complexes. These complexes link transmembrane protein cargoes to clathrin. They mediate intracellular membrane trafficking between the trans-Golgi-network (TGN), endosomes, lysosomes, and the plasma membrane [148,149]. Each AP complex consists of four different subunits: For AP-1 these are γ, β1, μ1, and σ1; and for AP-2, they are α, β2, μ2, and σ2 [150,151,152,153]. The AP complexes bind short-linear motifs such as YxxΦ (tyrosine-based motifs) and [DE]xxxL[LI] (acidic leucine-based motifs) in the cytoplasmic domains of their cargos [152]. YxxΦ motifs bind to µ subunits, whereas [DE]xxxL[LI] motifs bind primarily to σ subunits with contributions to the binding site from adjacent subunits (α in AP-2 and γ in AP-1). Acidic clusters (often containing phosphoserines—PSAC motifs) in the cytoplasmic domains of cargo proteins also bind the µ subunits, whose surfaces are basic [106,154,155]. Nef, Vpu, and their targets utilize all of these mechanisms to bind AP complexes. Both viral proteins can either interact with µ subunits via their acidic clusters (not shown here for Vpu but see Figure 5C for Nef), or they can interact via their acidic-leucine based motifs with the α/σ2 or γ/σ1 hemi-complexes (Figure 6A–E) [30,90,106,108,113,154]. Why these different modes of interaction with AP complexes? In the case of Nef, the different modes relate to the modulation of different targets. To modulate MHC-I, Nef interacts via its acidic cluster (plus other sequences) with µ1. In contrast, to modulate CD4, Nef interacts via its acidic leucine-motif with α-σ2. Whether this principle of “different modes for different targets” applies to Vpu remains to be determined. How Vpu and Nef take advantage of sequences in their different targets also seems target-specific. To link MHC-I to AP-1, Nef enables the tyrosine in the sequence YSQA in the MHC-I α chain to use the tyrosine-binding pocket on µ1, even though that sequence lacks a critical hydrophobic residue at the Y+3 position [106]. Nef accomplishes this by forming a ternary interaction interface involving itself, the cytoplasmic domain of MHC-I, and µ1 (Figure 5). In contrast, to link BST-2 to AP-1, Vpu allows the cytoplasmic domain of BST-2 to bind to µ1 via its YxxΦ sequence, while binding to the σ1-γ subunits of AP-1 via its ExxxLV sequence (Figure 6D–E) [30]. Thus, the interaction between the TMDs of Vpu and BST-2 (reviewed above) brings together the interaction of each protein’s cytoplasmic domains with AP-1, presumably increasing the net affinity and altering the trafficking pathway of the protein-complex. CD4 and Nef exemplify yet another mechanism. Both Nef and CD4 contain leucine-based motifs, and each can bind similarly to the α-σ2 subunits of AP-2 [108,113]. To bind AP-2 on its own, serines in the cytoplasmic loop of CD4 must be phosphorylated, providing a negative charge upstream of the leucines in the sequence SQIKRLL. This enables the sequence to behave as if it were an acidic leucine-based motif. In contrast to CD4, Nef binds AP-2 constitutively: The C-terminal loop (residues 149–179) of Nef contains an acidic leucine-motif required for modulation of CD4, and this motif and the entire loop become well-ordered when bound to α-σ2 (Figure 6A–C). Although modulation by Nef does not require the serine in the SQIKRLL of CD4, it does require the isoleucine and leucine residues. These observations lead to a “connector” model in which the cytoplasmic domain of CD4 binds Nef, presumably via a hydrophobic interaction, while Nef binds AP-2. Exactly how CD4 and Nef interact, as discussed above, remains uncertain.

## 5. What Else Is Known about “Global” Modulation of the Plasma Membrane by the Virus and What Is the Significance of These Cellular Targets?

### 5.1. Downregulated Proteins

Many proteins antagonized by Vpu and Nef have been identified, mostly by non-systematic approaches (Table 1). These include CD4, MHC-I, BST-2, and SERINC5 as described above, as well as homing receptors (CCR7, CD62L), co-receptors required for viral entry (CCR5, CXCR4), cell surface proteins associated with NK/NKT or T cell activation (NTB-A, PVR, CD1d, NKG2D, CD28, TIM-1) and adhesion molecules (ICAM-1). Most of these are restriction factors or immuno-receptors plausibly linked to viral replication or immune evasion. What “global” discovery approaches have been used to identify new targets of Nef and Vpu, and what has been learned from them? A screening study of cell surface molecules using available antibodies to over 100 proteins found that more than 30% were modulated by Nef, and most of those were also modulated by Vpu, including members of the tetraspanin family [156]. An unbiased, systems-level approach combining plasma membrane enrichment (cell surface biotinylation) with quantitative mass-spectrometry provided a comprehensive catalogue of how HIV-1 remodels the T cell surface [157]. Over 100 plasma membrane proteins were identified as depleted from the plasma membrane by HIV-1 infection, including well-known targets of Vpu and Nef (CD4, BST-2, SERINC3, and SERINC5) as well as, among others, SNAT1, an alanine transporter targeted by Vpu. Depletion of this transporter from the plasma membrane inhibits cell division and consequently T cell activation, but how this benefits a virus that replicates best in activated cells is unclear. Given the large number of proteins modulated by Vpu and Nef, the questions of whether some of them are incidental or “off”-targets, or whether some are modulated as part of a complex with other proteins, remain open.

### 5.2. Significance

One approach to address significance in screening assays is to bias the search in favor of proteins with a high predetermined likelihood of biological importance, for example, interferon stimulated gene products (ISGs). A relatively high-throughput platform (Global Arrayed Protein Stability Analysis; GAPSA) was used to search for ISG products that are degraded by Vpu [158]. Using a cDNA library composed of over 400 ISGs, new Vpu degradation targets including the transmembrane proteins CD99 and PLP2 were identified. CD99 was also identified as a Vpu-target in an independent quantitative proteomic study of plasma membrane proteins modulated by Vpu (along with ICAM-1 and -3) [159]. Both CD99 and PLP, when ectopically expressed, inhibited virion-infectivity in a manner partially rescued by Vpu [158].

Another approach to assessing the significance of the cellular targets of Vpu and Nef is genetic. Viral pathogens appear to be major drivers of evolutionary change in the human proteome [160]. Genetic evidence of an evolutionary arms race between hosts and their viruses—positive selection detected by high ratios of synonymous to non-synonymous mutations among the coding sequences of primate or mammalian genes—has been found for most antiviral restriction factors, including BST-2, SAMHD1, APOBEC3G, and TRIM5α [145,161,162,163,164]. Presumably, the viral proteins that interact with these factors drive this diversity. Table 1 shows several examples of functional redundancy between Vpu and Nef in the case of HIV-1 and human proteins. But not shown is the striking manner in which the activities of Vpu and Nef have toggled back and forth across primate evolution when simian immunodeficiency viruses (SIVs) and their host orthologous genes encoding BST-2 are considered. Most SIVs use Nef rather than Vpu to antagonize BST-2 [165,166]. Moreover, SIV Nef cannot antagonize human BST-2, nor can HIV-1 Vpu antagonize simian BST-2. These specificities map to regions of BST-2 that are under positive selection, specifically the TMD required for the interaction with Vpu and the cytoplasmic domain required for antagonism by Nef [145,162]. As noted above, the SERINC proteins differ from BST-2 and most others restriction factors in that they do not show genetic evidence of an ongoing arms race with pathogens at the gene-level, despite their biological interaction with Nef proteins of HIVs and SIVs [51]. Nonetheless, a few codons within SERINC3 and SERINC5 appear to be under positive selection. In SERINC3, two of these residues are serines that are part of an acidic cluster able to direct binding to the μ subunits of the clathrin adaptors AP-1 and AP-2 [154]. These residues coordinately toggle between serine, asparagine, and glycine across mammalian evolution, suggesting that their ability to bind clathrin adaptors varies. This suggests that analysis of genetic selection can identify not only protein–protein interfaces between restriction factors and their targets or antagonists but also the interfaces between restriction factors and the cellular co-factors that support antagonist-activity.

Why *serinc3* and *serinc5* do not show evidence of an evolutionary arms race at the gene-level, despite their broad antiretroviral activity, which includes genetically distant retroviruses of different species (HIV-1 and SIVs of primates, murine leukemia virus of mice, and equine infectious anemia virus of horses, all of which encode SERINC antagonists), remains to be determined.

Finally, the extent to which the analysis of genetic selection might be applied to high-depth proteomic studies to winnow out bystander or off-target effects and enable focus on the “real” restriction factors antagonized by Vpu and Nef is an open question. For example, are tetraspanins [167,168]; Tim proteins, which inhibit virion release by binding phosphatidylserine and are counteracted by Nef [169]; or amino acid transporters under positive selection [157]? Answering these questions requires attention to the specific codons of the protein and whether those under positive selection are consistent with the mechanisms of antagonism described above. For example, human CD4 is under positive selection at the gene-level, but none of the individual codons under selection are in the transmembrane or cytoplasmic domains of the protein [170], the regions that support modulation by Vpu and Nef.

## 6. Conclusions

Modulation of the protein content of the plasma membrane is a key aspect of how HIV-1 adapts the host environment to its needs. The importance of this modulation is underscored by the virus’s dedication of two of its nine genes (*vpu* and *nef*) to this purpose. The consequences for the virus are multiple: Protection of the infected cell from cellular immunity by down-regulation of class I MHC and of specific receptors for natural killer cells, counteraction of the cell-intrinsic restriction factors BST-2 and SERINC5, and avoidance of the deleterious consequences of CD4-Env interaction in the virus-producer cell. The Vpu and Nef proteins have many differences including their timing of expression, membrane-topology, and modes of interaction with their cellular targets. Yet, they share common targets (particularly CD4) as well as some mechanisms of action (modulation of the specificity of clathrin adaptors to target cellular proteins toward lysosomal degradation). While certain mechanistic aspects of Vpu- and Nef-activity are well known, others remain to be elaborated, in particular the structure of key interfaces between these viral proteins and their cellular targets and cofactors. The plasma membrane-associated restriction factors reviewed in depth here, BST-2 and SERINC5, are only two of the many cellular proteins whose residence on the plasma membrane is affected by Vpu or Nef. Sorting out the roles of these additional plasma membrane proteins, in particular defining the extent to which their modulation contributes to viral fitness, is a current challenge to the field.

## Figures and Tables

**Figure 1 cells-08-01020-f001:**
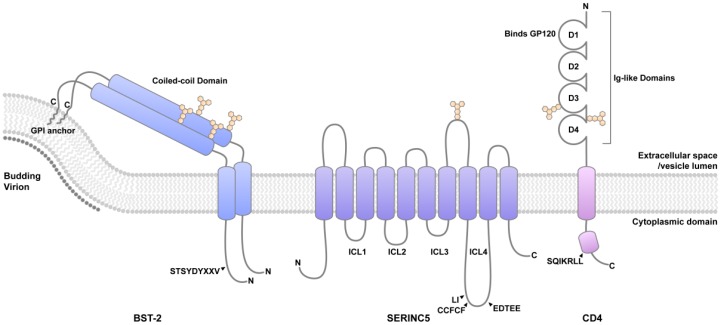
The cellular proteins BST-2, SERINC5, and CD4. BST-2 has a short cytoplasmic domain and two membrane anchors, a transmembrane (TM) domain, and a glycosyl-phosphatidylinositol (GPI) anchor, separated by an extracellular coiled-coil. This topology enables BST-2 to partition one end—usually the GPI anchor—into the lipid bilayer of the budding virion while the other end remains in the plasma membrane. The short cytoplasmic domain of BST-2 contains sites for ubiquitination (STS), and a clathrin mediated endocytic motif, YxYxxV. The BST-2 ectodomain has two N-linked glycosylation sites, shown in tan color. SERINC5 is a multi-pass transmembrane protein containing 10 transmembrane domains, 5 extracellular loops, and 4 intracellular loops. The single N-linked glycosylation site is indicated. ICL4 contains determinants of sensitivity to Nef: L350 and I352 and a palmitoylation motif, CCFCS, support Nef-responsiveness, whereas the EDTEE sequence, which binds clathrin adaptor proteins, seems to inhibit it. CD4 is a glycosylated, dimeric integral membrane protein belonging to the immunoglobulin superfamily. It has four extracellular Ig domains (D1–D4); domain D1 interacts with HIV envelope glycoprotein. CD4 contains two glycosylation sites. The membrane proximal cytoplasmic region is α-helical and contains hydrophobic residues (I410, L413, L414) that facilitate the interaction with clathrin adaptor proteins and are required for Nef-mediated down-regulation.

**Figure 2 cells-08-01020-f002:**
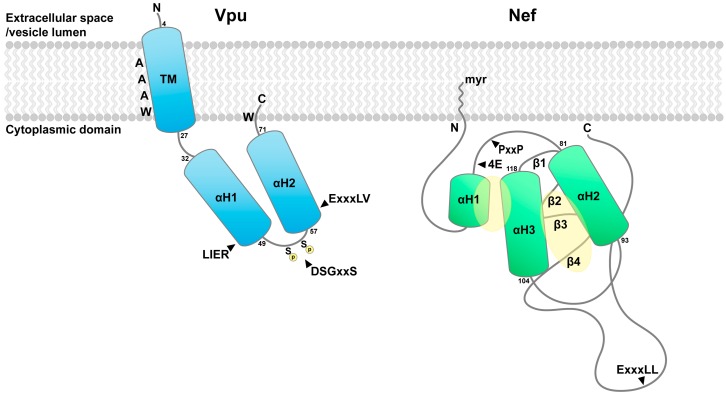
The viral proteins Vpu and Nef. Vpu (left) is a small, type-I transmembrane protein. The transmembrane (TM) α-helix displays an alanine-face, which interacts with the transmembrane helices of some of Vpu’s cellular targets. The cytoplasmic domain, when not bound to any cellular partner, contains two α-helices; between them is a DSGxxS motif. When the serines are phosphorylated (P in yellow circle), this motif supports binding to β-TrCP, linking Vpu to a multi-subunit E3 ubiquitin ligase, and to the µ subunits of AP-1 and AP-2, linking Vpu to clathrin. The LIER sequence supports binding to β-TrCP. The ExxxLV sequence supports a second mode of binding to the clathrin adaptor AP-1 (see Figure 6). Tryptophan residues (W) anchor the TM domain, and in some clades of HIV-1 attach the C-terminus to the lipid bilayer. Nef (right) is a small, peripheral membrane protein. It associates with membranes via N-terminal myristoylation (myr). The N-terminus of Nef (up to the PxxP region) is conformationally flexible; the schematic shown represents a putative conformation associated with the modulation of CD4, in which helix1 (H1) including residues W57 and L58 (not indicated) interacts with helix3 (H3). This interaction forms a potential binding site for the cytoplasmic domain of CD4 (yellow, transparent ellipse). The acidic cluster (4E) supports binding to the µ subunit of AP-1 and is required for the modulation of class I MHC, as is the PxxP region (see Figure 5). In addition to forming the binding pocket for the cytoplasmic domain of the MHC-I α-chain, the PxxP region forms a binding region for the SH3 domains of Src-family kinases that contribute to the modulation of MHC-I [118]. The Nef “core” contains two α-helices (H2 and H3) and a network of β-stands (β1–β4). The “upper” aspect of the cleft between helices H2 and H3 forms part of the SH3-binding domain, whereas the “lower” aspect forms an unfilled hydrophobic pocket (yellow, transparent ellipse). This pocket is an alternative binding site for the cytoplasmic domain of CD4; it could bind the ICL4 of SERINC5; or it could participate in binding the AP complexes. The ExxxLL motif of Nef is within a loop that emerges from the β-stand network. This motif binds in a canonical manner to the σ and large specific subunits of AP-1 and AP-2 (see Figure 6 for the interaction with AP-2). The ExxxLL motif is required for the modulation of CD4 and SERINC5 but not for the modulation of MHC-I.

**Figure 3 cells-08-01020-f003:**
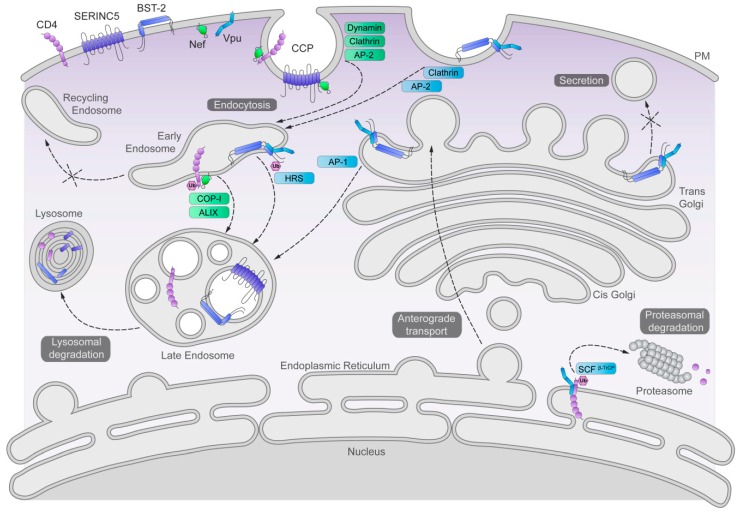
Cell biologic schematic of protein quality control and membrane trafficking pathways co-opted by Vpu and/or Nef. Vpu and Nef, their targets—CD4, SERINC5, and BST-2—and their co-factors—the SCF-E3 ubiquitin ligase, clathrin, AP-1, AP-2, Dynamin, HRS, ALIX, and β-COP—are indicated. Complexes of Vpu with CD4 or BST-2, and Nef with CD4 or SERINC5, are shown. Arrows indicate direction of transport. CCP: clathrin-coated pit; PM: plasma membrane; Ub, ubiquitin. Details of the illustrated pathways are in the text.

**Figure 4 cells-08-01020-f004:**
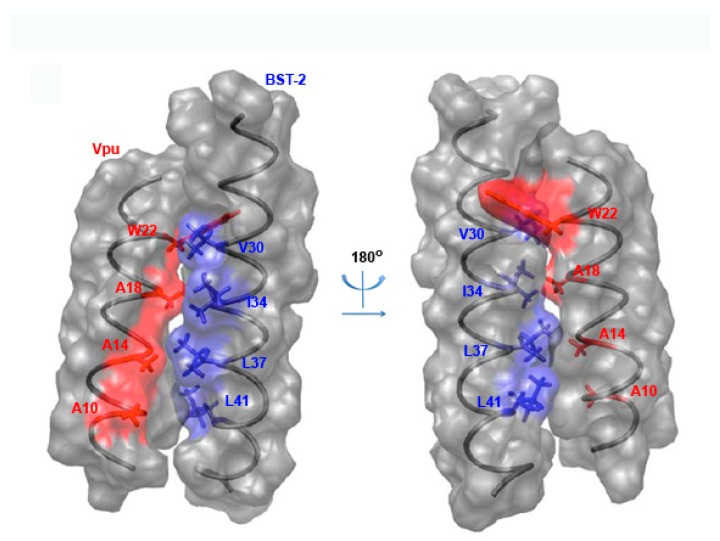
Structural interface between Vpu and BST-2. The Vpu and BST-2 transmembrane helices are shown in an antiparallel orientation. On the Vpu-side, the interface consists of an alanine-face (A10-A14-A18) followed by a membrane-anchoring tryptophan (W22), which likely inserts its side chain among the lipid head groups on the cytoplasmic side of the bilayer. On the BST-2-side, hydrophobic residues (L41-L37-I34) project their side chains toward the alanine-face of Vpu.

**Figure 5 cells-08-01020-f005:**
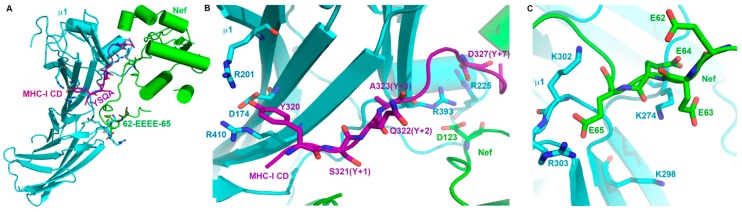
Structural interface between Nef, the MHC-I α chain cytoplasmic domain (CD), and the medium subunit of AP-1 (µ1). (**A**) Crystal structure of the ternary complex of Nef-MHC-I-αCD-µ1 (PDB:4EN2). Nef is shown in green; MHC-I-αCD is shown in magenta; and µ1 is shown in cyan. The membrane-proximal end of MHC-I-αCD containing the YSQA sequence and the acidic cluster of Nef are indicated. (**B**) Interaction of MHC-I-αCD YSQA sequence with the tyrosine-binding pocket residues (R201, D174, R410) of µ1 is shown; and the ternary interaction of D327 of MHC-I-αCD with Nef D123 and µ1 basic residues R225, R393 is also shown. (**C**) Interaction of Nef’s acidic cluster (62-EEEE-65) with µ1-basic residues (K274, K298, K302, R303).

**Figure 6 cells-08-01020-f006:**
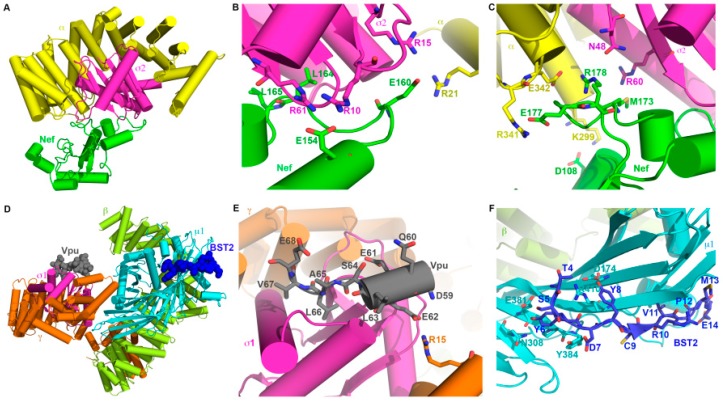
Structural interfaces between Nef and AP-2 (A–C) and Vpu, BST-2, and AP-1 (D–F). (**A**) Crystal structure of Nef bound to a hemi-complex of AP-2 containing two of the four AP subunits, α and σ2 (PDB: 4NEE). α is shown in yellow; σ2 is shown in magenta; and Nef is shown in green. (**B**) Interaction of Nef’s acidic leucine motif with α-σ2. Nef L164 and L165 interact with σ2 (magenta); Nef E160 interacts with R15 of σ2 (magenta) and R21 of α (yellow); Nef E154 interacts with R10 and R61 of σ2 (magenta). (**C**) Interaction of Nef distal C-terminal loop (173-178) with α- σ2. Nef M173 interacts with R60 of σ2 (magenta); Nef E177 interacts with R341 of α; and Nef R178 interacts with E342 of α (yellow). (**D**) Crystal structure of the Vpu/BST2/AP-1 complex (PDB: 4P6Z). All four subunits of the AP-1 complex are shown. Vpu (black) and the BST2 cytoplasmic domain (blue) are shown by spheres; only a small portion of the Vpu cytoplasmic domain is resolved. AP-1 subunits are colored: σ1 (magenta), γ (orange), β1 (green), and µ1 (cyan). (**E**) Interaction of Vpu’s acidic leucine motif with γ-σ1. Vpu residues are shown in black sticks and labeled; the key residues are E62, L66, and V67. This interaction is analogous to that of the Nef acidic leucine motif with α-σ2 shown in panel B. R15 of γ is shown by orange stick. (**F**) Interaction of the BST-2 cytplasmic domain with the medium subunit of AP-1 (µ1). BST-2’s tyrosine motif—Y6xY8xxV—binds in a pocket of µ1. BST-2-Y8 interacts with D174 and R410 of µ1, the canonical binding site for tyrosines within YxxΦ motifs.

**Table 1 cells-08-01020-t001:** Partial Summary of Plasma Membrane Proteins Modulated by HIV-1 Vpu and Nef.

										Downregulation/Degradation		Binding		
		Modulated by		Cellular Adaptor Utilized by Vpu or Nef			Counteraction Mechanism			Domains or Residues Required		Domains or Residues Required		
Host PM Protein	Biological Function	Nef	Vpu	AP-1	AP-2	SCFb-TrCP	Proteasomal Degradation	Lysosomal Degradation	Endosomal Sequestration	Nef	Vpu	Nef	Vpu	Reference(s) (PMID#)
BST-2 (CD317)	ISG: Traps enveloped viral particles on plasma membrane		✓	✓	✓	✓		✓	✓	N/A	TM (A10,A14,18,W22), S52,56; 59ExxxLV	N/A	TM (A10,A14,18,W22)	18200009; 18342597
CCR5 (CD195)	Chemokine receptor: inflammatory response	✓						✓	✓	G2; 62EEEE; PxxP		Unknown	N/A	15854903; 26178998
CCR7 (CD197)	Homing receptor: recruitment of immune cells to lymphoid tissues		✓						✓	N/A	TM (A10, A14, A18, W22)	N/A	Unknown	24910430
CD1d	APC: Present lipid antigens to NKT cells	✓	✓						✓			Unknown	CD (APW76) cladeB	15916790; 16385629; 20530791; 25872908
CD28	TCR complex: co-stimulation, activation	✓	✓					✓		LL165; DD175	59ExxxLV; S52,56	DD175	S52,56	29329537
CD4	TCR co-receptor: T cell activation; HIV-1 Env receptor	✓	✓		✓	✓	✓	✓		G2; 57WL; G95; G96; L97; R106; L110; 160ExxxLL; 174DD	TM; L63; V68; S52, 56	57WL, G95, G96,L97, R106, L110	TM; CD	3118220; 1433512
CD62L	Leukocyte adhesion and signaling	✓	✓						✓	Unknown	Unknown	Unknown	Unknown	25822027
CXCR4 (CD184)	Chemokine receptor: inflammatory response	✓							✓	62EEEE; PxxP		Unknown	N/A	16928758
ICAM-1 (CD54)	Leukocyte adhesion; NK cell activation		✓			✓	✓			N/A	TM (A10,A14, A18); S52,56		TM (A10,A14, A18)	28148794
MHC-I	Induction of Adaptive Immunity (CD8+ T cells): Antigen presentation to APCs	✓ HLA-A/B	✓ HLA-C		✓			✓		W13; R17; R19; M20; 62EEEE; P78; W113; Y120; D123	TM (LE5; L16; L18 in WITO)	Tri-molecular complex with AP-1 (W13, M20, 62EEEE, P78, D123)	Unknown	8612235; 9450757; 22705789; 27173934
MHC-II	Induction of Adaptive Immunity (CD4+ T cells): Antigen presentation to APCs	✓								62EEEE; P75; P78; LL164,165	N/A	N/A		11593029
NKG2D-L	Activation receptor: induction of NK cell mediated cytotoxicity and cytokine release	✓								G2	N/A	N/A		17170457; 19424050
NTB-A (CD352)	Co-activation receptor: induction of NK cell mediated cytotoxicity and cytokine release		✓						✓	N/A	TM (A18)	N/A	TM	21075351
PVR (CD155)	Activation receptor: induction of NK cell mediated cytotoxicity and cytokine release	✓	✓	✓					✓	72PxxPxxP; 62EEEE; F191	TM (A10,A14, A18), S52,56	Unknown	TM (A10,A14, A18)	22301152; 25113908
SERINC3/5	Phospholipid biosynthesis; Reduce retroviral infectivity	✓			✓			✓	✓	G2; CAW57; D123; LL165; ED175		I109; L112; W115; F121		26416734; 29514909; 27681140
SNAT1	Immunometabolism (amino acid - alanine - transporter)		✓			✓		✓			S52,56; TM (W22)		Unknown	26439863
Tetraspanins	Membrane organization	✓	✓					✓		Variable	S52,56; TM (partial)	Unknown	Unknown	25275127; 25568205
CD99	PM T cell receptor: Regulator of focal adhesions, cell-cell junctions		✓							N/A	Unknown	Unknown	Unknown	29490283
PLP2	Membrane trafficking		✓							N/A	Unknown	Unknown	Unknown	29490283
TIM-1 (CD365)	T-cell activation, cellular proliferation, apoptosis, immune tolerance	✓			✓				✓	G2; D123, LL165	N/A	Unknown	Unknown	30842281

AP-1/2: Adaptor Protein 1/2; APC: Antigen Presenting Cell; ISG: Interferon Stimulated Gene; NK: Natural Killer; NKT: Natural Killer T cell; PM: Plasma Membrane; PMID: PubMed identifier number; TM: Transmembrane domain; CD: cytoplasmic domain; SCFb-TrCP: Skp1/Cullin1/F-box ubiquitin ligase complex containing b-TrCP; PxxP: polyproline region; WITO: Vpu derived from primary transmitted founder HIV-1 clone; HLA: human leukocyte antigen.

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
