# Peer review of "Plasma Membrane-Associated Restriction Factors and Their Counteraction by HIV-1 Accessory Proteins"

_cells, 2019, doi:10.3390/cells8091020_

Round 1

Reviewer 1 Report

Ramirez et al provide an excellent review of Vpu and Nef function in the context of BST-2, SERINC5 and CD4.  In contrast, the modulation of HLA class-I molecules is given more perfunctory mention.  The mechanism of class-I downregulation by Nef is described, but not intriguing recent findings that Nef and Vpu target non-overlapping subsets of HLA class-I molecules, with transmembrane domains residues responsible for the latter interaction [27173934,30180214].  This new data should be included given the importance of class-I molecules in HIV control in vivo. 

Author Response

The reviewer makes a very good suggestion; thank you. We have added new text to note the specificity of Nef (lines 294-296) and Vpu (lines 265-266) with respect to modulation of HLA, including noting the role of residues in the Vpu  transmembrane domain (lines 265-266). We also specified the specificity of Nef (HLA-A and -B) and Vpu (HLA-C) in the Table.

Reviewer 2 Report

Ramirez et al. picked up two accessory proteins of HIV, vpu and nef, sharing similar function to downregulate host proteins on the plasma membranes. Main topics of this review are similar to those of the previously published review (Sugden et al. Viruses 2016), but the contents of the present manuscript are well organized and comprehensive.

Major suggestions;

It is better to explain a bit more about ILT-7 in line 115 focusing on immune-sensing. In addition, ILT-7 should be spelled out in full before use abbreviation. It is better to add the discussion on MHC class I down-regulation by nef. Specifically, it should be noted that nef down-regulates HLA-A and B but not C. This selective downregulation is important in escape from NK cells, so the mechanism how nef distinguishes HLA-C should be discussed.

Minor points;

In the figure legend of Fig.1 in line 137, D4 should be D1. In line 139, 413L and 414L must be L413, and L414.

Author Response

The reviewer has made very good points; thank you.

Regarding ILT7, we spelled this out, and added a reference regarding the direct interaction between the two proteins (lines 114-121).

Regarding Nef and class I MHC (HLA), we added that Nef modulates HLA-A and -B but not -C, and explained why mechanistically (lines 294-296). We also added this information to the Table.

We made the minor corrections to the legend of Figure 1 (lines 138 and 140).